# Pyrolysis-GCMS of *Spirulina platensis*: Evaluation of biomasses cultivated under autotrophic and mixotrophic conditions

**Sueilha F. A. Paula**[1]☯, **Bruna M. E. Chagas**[2]☯*, **Maria I. B. Pereira**[3]‡, **Adriano H. N. Rangel**[3]‡, **Cristiane F. C. Sassi**[4]‡, **Luiz H. F. Borba**[3]‡, **Everaldo S. Santos**[5]‡, **Estefani A. Asevedo**[5]‡, **Fabiana R. A. Câmara**[3]‡, **Renata M. Araújo**[1]☯

1 Institute of Chemistry, Federal University of Rio Grande do Norte, Natal, State of Rio Grande do Norte, Brazil, 2 Infrastructure Superintendence, Federal University of Rio Grande do Norte, Natal, State of Rio Grande do Norte, Brazil, 3 Specialized Academic Unit in Agrarian Science, Federal University of Rio Grande do Norte, Natal, State of Rio Grande do Norte, Brazil, 4 Department of Systematic and Ecology, Federal University of Paraíba, João Pessoa, State of Paraíba, Brazil, 5 Department of Chemical Engineering, Federal University of Rio Grande do Norte, Natal, State of Rio Grande do Norte, Brazil

☯ These authors contributed equally to this work.
‡ These authors also contributed equally to this work
* brunam.emerenciano@gmail.com

**Data Availability Statement:** All relevant data are within the paper and its Supporting information files.

## Abstract

Microalgae are autotrophs and $CO_2$ fixers with great potential to produce biofuels in a sustainable way, however the high cost of biomass production is a challenge. Mixotrophic growth of microalgae has been presented as a great alternative to achieve economic sustainability. Thus, the present work reports the energetic characterization of *S. platensis* biomasses cultivated under autotrophic (A) and mixotrophic conditions using cheese whey waste at different concentrations, 2.5 (M2.5), 5.0 (M5) and 10.0% (M10), in order to analyze the potential production of valuable chemicals and bio-oil by TGA/DTG and Py-GC/MS. The biochemical compositions of the studied biomasses were different due to the influence of different culture mediums. As the whey concentration increased, there was an increase in the carbohydrate content and a decrease in the protein content, which influenced the elemental composition, calorific value, TGA and volatile compounds evaluated by Py-GC/MS at 450˚C, 550˚C and 650˚C. Sample M10 had lower protein content and formed a smaller amount of nitrogenates compounds by pyrolysis at all temperatures evaluated. There was a reduction of 43.8% (450º), 45.6% (550ºC) and 23.8% (650ºC) in the formation of nitrogenates compounds in relation to sample A. Moreover, the temperature also showed a considerable effect in the formation of volatile compounds. The highest yields of nitrogenates compounds, phenols and aromatic and non-aromatic hydrocarbons were observed at 650ºC. The oxygenated, and N and O containing compounds decreased as the temperature increased. Hydrocarbons such as toluene, heptadecane and heneicosane were produced by *S.platensis* pyrolysis, which makes this biomass attractive for production of high quality bio-oil and valuable chemicals. Therefore, the results showed that it is possible to decrease the formation of nitrogen compounds via manipulation of growth conditions and temperature.

**Funding:** The author(s) received no specific funding for this work.

**Competing interests:** The authors have declared that no competing interests exist.

## Introduction

The high demand for fuels associated with the decrease of fossil fuel future shortages has been stimulating the search for new energy sources. Additionally, combustion of fossil fuels produce large quantities of carbon dioxide, contributing to the enhanced greenhouse effect [1]. Thus, microalgae has been presented as a promising alternative biomass to produce biofuel and $CO_2$ capture, since they offer high growth rate when compared to higher plants, cultivation versatility and adaptation to climatic variations, $CO_2$ consumption through photosynthesis minimizing the greenhouse effect. In addition, the cultivation of microalgae can be carried out using waste from agro-industry, urban sewage, brackish and/or saline water, among others [2].

Microalgae biomass are important to produce biofuel including bio-oil, biogas, bioethanol, and biodiesel. However, the high cost of producing biomass limits large-scale applications [3]. According to Xia and Murphy [4], inorganic growth medium use can reach up to 50% of microalgae cultivation cost. Thus, microalgae mixotrophic cultures have been shown as a promising strategy to reduce this cost. In this process, wastes rich in organic carbon can be assimilated by microalgae being a capable alternative of providing economic viability to the process [5]. Due to the ability to assimilate carbon sources, microalgae under mixotrophic cultivation conditions are less dependent on photosynthesis so cell growth is not limited by photoinhibition leading to higher biomass productivity [6].

Pyrolysis is a thermal degradation process of biomass capable of converting biomass into gaseous, liquid and solid biofuels and valuable chemicals. The liquid product, called bio-oil, has a high-energy content with great potential to replace diesel [7]. The obtained results show that microalgae and proteinaceous biomass bio-oil has better chemical properties than lignocellulosic bio-oil, being more stable, presenting lower oxygen content and higher calorific value [8,9]. These properties are justified by conversion of protein and lipids fraction into aromatic and linear hydrocarbons, respectively [10,11].

*S. platensis* bio-oil produced through thermal conversion processes has been studied due to its potential [9,12–14]. Moreover, the *S. platensis* is a high-protein cyanobacterium species that has a high photosynthetic capacity and produces large amounts of biomass per unit area when compared to lignocellulosic and oilseed biomasses [15]. Despite the advantages, pyrolysis of protein-rich biomass, including microalgae, produces a bio-oil with a high nitrogen content, reaching up to more than 10% of elemental content. This limitation compromises the industrial pyrolysis process since this bio-oil cannot be used directly as fuel due to the large amounts of $NH_3$, HCN and $NO_x$ that could be released during combustion [16]. However, compounds present in bio-oil such as pyrrole, pyridine, and indole could have important applications for pharmaceuticals and chemical industry [17].

The N transformation behavior in the process is still not fully understood and studies involving strategies for the production of bio-oil with low N content are required [7]. One of the techniques used is denitrogenation via bio-oil upgrading, however these are processes expensive and poorly understood. One of the ways to produce microalgae bio-oil with less nitrogen content can be via a biomass with less protein content, once that the bio-oil composition follows the trend of the biomass elemental composition. It has been shown that microalgae under mixotrophic growth conditions reduce protein synthesis and produce more carbohydrates [18–21]. Thus, mixotrophic growth can be a sustainable strategy to produce *S. platensis* biomass with a lower protein content and, consequently, form a smaller amount of N-containing compounds during pyrolysis.

Pereira et al. [18] cultivated *S.platensis* under autotrophic and mixotrophic conditions using cheese whey as organic carbon source. For the mixotrophic cultures, the Zarrouk medium was supplemented with 2.5%, 5.0%, and 10% buffalo mozzarella cheese whey [18].

When compared with the photoautotrophic culture, mixotrophic microalgae grew faster, producing higher biomass yield and higher carbohydrate content in a short period of time, but they show a decrease in protein and lipid productivity. The protein contents were 65.55% for autotrophic conditions and 60.62, 44.56 and 39.62% for mixotrophic conditions with 2.5, 5.0 and 10.0% of whey, respectively. It is possible to observe a direct relationship between the increase of whey percentage and the decrease of total protein synthesis for all the studied conditions.

In this regard, the *S.platensis* biomasses with different protein contents obtained by Pereira *et al.* [18] were energetically characterized and used to study the thermal degradation by TGA and Py-GC/MS as a strategy to evaluate the production of volatile compounds at temperatures of 450, 550 and 650˚C. This study explores *S. platensis* biomass cultivated under autotrophic and mixotrophic conditions using whey as a waste from dairy industry that could be a strategy to produce biomass with lower cost associated with treating wastewater and sequestering $CO_2$. In addition, the biomass can be thermally converted via pyrolysis to produce bio-oil and valuable chemicals.

## Materials and methods

### Biomass

The *S.platensis* biomasses were obtained under autotrophic and mixotrophic growth conditions by Pereira *et al.* [18] and they provided the feedstock to carry out the experiments of this work. The *S.platensis* strain was collected from the Microalgae Bank of the Laboratory of Environments and Biotechnology with Microalgae (LARBIM), Federal University of Paraíba. The microalgae was maintained under sterile conditions using standard culture medium [22] (Table 1). For the mixotrophic cultures, the Zarrouk medium was supplemented with 2.5%, 5.0%, and 10% of buffalo mozzarella cheese whey from TAPUIO Agropecuária Ltda cheese industry located in Rio Grande do Norte, in Northeast region of Brazil. The whey was clarified for cultures and the precipitated material was removed during this process. The clarification process involves high temperatures causing the precipitation of coagulated proteins. Pereira *et al.* [18] demonstrated that the content of total solids in the clarified whey was 6.77% being 0.02% of fat, 0.60% of protein and 5.07% of lactose. Cultures were performed aseptically with a

**Table 1. Composition of the Zarrouk Component Quantity medium (g.L$^{-1}$) (Zarrouk [22]).**

| | |
|---|---|
| NaHCO$_3$ | 16.8 |
| K2HPO$_4$ | 0.5 |
| NaNO$_3$ | 2.5 |
| K2SO$_4$ | 1.0 |
| NaCl | 1.0 |
| MgSO$_4$.7H$_2$O | 0.2 |
| CaCl$_2$ | 0.04 |
| FeSO$_4$.7H$_2$O | 0.01 |
| EDTA | 0.24 |
| A5* | 1.0 mL.L-1 |
| B6** | 1.0 mL.L-1 |

A5 Solution composition A5: 2.86 g/L of H$_3$BO$_3$; 1.81g/L of MnCl$_2$.4H$_2$O; 0.222g/L of ZnSO$_4$.7H$_2$O; 0.079g/L of CuSO$_4$.5H$_2$O; 0.015g/L of NaMoO$_4$.

**B6 Solution composition: 22.96 mg/L of NH$_4$VO$_3$; 96 mg/L of K$_2$Cr$_2$(SO$_4$)$_3$.24H$_2$O; 47.85 mg/L of NiSO$_4$.7H$_2$O; 17.94 mg/L of Na2WO$_4$.2H$_2$O; 61.1 mg/L of TiOSO$_4$.H$_2$SO$_4$.8H$_2$O; 43.98 mg/L of CO(NO$_3$)$_2$.6H$_2$O.

luminous intensity of 238 μmol m$^{-2}$ s$^{-1}$, 12h light/dark photoperiod under constant sterile aeration. The biomasses were collected in the stationary phase by filtration, then, the biomass was lyophilized and frozen at -20˚C until the time of the chemical analysis. More details of growth conditions can be seen in Pereira *et al.* [18]. In order to facilitate the organization of this work, the samples were represented by the following acronyms: A (biomass obtained from autotrophic cultivation) and M2.5, M5 and M10 for biomasses cultivated under mixotrophic conditions with addition of 2.5, 5.0 and 10% of buffalo mozzarella cheese whey, respectively.

## Characterization of *S.platensis*

Pereira *et al.* [18] previously analyzed protein, moisture, ash content, carbohydrate and lipid content of all samples utilized in this work, being the dry matter and ash content quantified according to AOAC [23]. The total protein content was determined by the Kjeldahl method according to 981.10 of the AOAC [24]. The conversion factor for protein content was 4.78, the amount recommended for microalgae [25]. Total carbohydrates determination was performed by the method of Korchet [26] and adapted by Derner [27]. The ultimate analysis was performed according to method ASTM D 3176 [28] using The Vario Micro Cube elemental analyzer, equipped with a thermal conductivity detector (TCD). The contents of carbon, hydrogen, nitrogen and sulfur were quantified, being the oxygen content determined by difference, %O = %C-%H-%N-%S-%H2O. The sample's calorific value determination was carried out using a Parr Instruments model 1341 calorimeter according to ASTM D 5468–02 [29] method.

## TGA

The samples were analyzed using a Shimadzu Thermogravimetric Analyzer (TGA) under an inert atmosphere using nitrogen gas with a flow rate of 50 ml min$^{-1}$. Before starting the assay, the samples were first crushed in order to obtain better homogenization and placed in previously identified alumina crucibles. The equipment used in the TGA is composed of a microbalance, an oven, a thermocouple and a flow system, being approximately 3 mg of the dry sample used for each analysis. The nitrogen gas is used as purge in order to eliminate the air in the reactor and to avoid the oxidation reaction of the samples. The samples were heated from 25˚C temperature to 600˚C, at a heating rate of 5˚C.min$^{-1}$.

## Fast pyrolysis

The fast pyrolysis of *S.platensis* biomasses cultured under autotrophic and mixotrophic conditions were performed through a CDS Analytical Pyroprobe 5200 microreactor equipped with a platinum resistance that can be heated to 1473 K at a maximum heating rate of 20 K/ms. All tests were performed in duplicate. For fast pyrolysis analysis, about 3.0 mg of the *S.platensis* lyophilized samples without pre-treatment was inserted into a quartz capillary using quartz wool (inert material) to avoid movement of sample inside the capillary as shown in Fig 1:

All samples were prepared using gloves to prevent contamination. The capillary was inserted in the platinum resistance and placed in the micropyrolyzer. Thus, to ensure that the sample would be inserted into the center of the pyrolysis chamber, the material was placed between the top and bottom of the quartz wool approximately 2.7 cm from the lower end of the tube. The samples were arranged in the autosampler vertically and moved by gravity into the pyrolysis chamber being purged and then pyrolyzed with the residence time of 10 s, heating rate of 20ºC/ms and temperature established in the method. The vapor from the thermal decomposition of the biomass was carried with helium 5.0 (99.999%) at a flow rate of 1 mLmin$^{-1}$ in the column and the heating rate used was 20˚Cmin$^{-1}$. The equipment is coupled

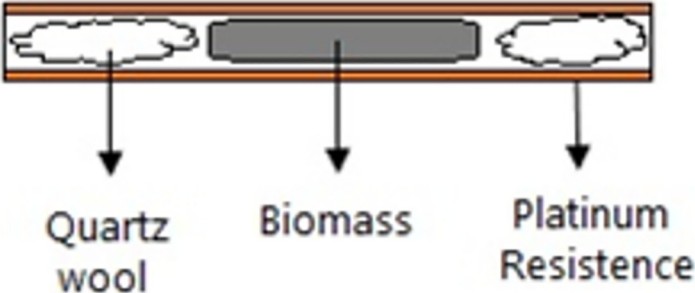

**Fig 1. Scheme of quartz capillary with quartz wool and biomass placed in the micropyrolyzer (Adapted from Simão *et al*. [11]).**

to a GC/MS QP2010 PLU gas chromatograph with a capillary column model Rtx-1701 (60mx0.25mmx0.25μm). During the analysis, the injector temperature was kept at 250°C and the interface temperature at 270°C, the ionization temperature was kept at 275°C. The split ratio was 1:90. The oven was programmed following an initial temperature pattern of 45°C for 4 min being heated to 280°C at a heating rate of 3°Cmin$^{-1}$. Data were processed using the NIST library and only components with 80% similarity were quantified since the compound content is considered linear with the area of the chromatographic peak [30].

## Statistical analysis

Statistical analysis was performed using the procedure of the Statistical Analysis System SAS® version 9.0. Experimental results were evaluated by Analysis of Variance (ANOVA) and Tukey's test, with a significance level of 5%. Data were analyzed using the PROC MEANS (to obtain descriptive statistics and check for possible data inconsistency) and PROC GLM (to analyze variance and test for comparison of means) of the SAS.
Where:
Dependent variable (Oxygen, Nitrogen, Oxygen/Nitrogen, Phenolic, Hydrocarbons and Aromatic HC) i-th treatment, in the j-th repetition;
Overall average;
Ti = Effect of the i-th treatment, where treatment is:
1) Class = 1 to 3 (450, 550 and 650)
2) Level = 1 to 4
εij = random error

## Results and discussion

### Composition of biomass

The composition of microalgae biomass varied depending on the cultivation conditions under programmed nutritional stress, once that these microorganisms are very versatile. The characterizations of the A, M2.5, M5 and M10 samples obtained under autotrophic and mixotrophic growth are shown in Table 2.

   The produced biomass, specifically for M5 and M10 showed an increase of 80.61% and 32.12% when compared to A, control culture. In this way microalgae under mixotrophy conditions have been shown a promising strategy due to its capacity of carbon assimilation from organic sources as well as conversion of light energy and $CO_2$ by photosynthesis. The increase of biomass production can be justified by a higher photosynthetic efficiency with less

Table 2. Characteristics of S. *platensis.*

| S. *platensis* biomass | % Buffalo mozzarella cheese whey | | | |
|---|---|---|---|---|
| | A | M2.5 | M5 | M10 |
| **Biomass Production (g.L$^{-1}$)** | | | | |
| | 1.65* | 2.16* | 2.98* | 2.18* |
| **Biochemical Analysis (wt.%) db** | | | | |
| Protein | 65.55* | 60.62* | 44.56* | 39.62* |
| Fat | 5.18* | 2.04* | 2.56* | 3.40* |
| Carbohydrate | 27.17* | 23.29* | 47.83* | 40.65* |
| Moisture | 6.35* | 8.23* | 9.43* | 5.08* |
| Ash | 6.04* | 14.21* | 14.03* | 6.36* |
| **Ultimate Analysis (wt.%) db** | | | | |
| C | 46.6 | 42.44 | 41.26 | 42.86 |
| H | 7.23 | 6.58 | 6.58 | 6.98 |
| N | 10.29 | 9.24 | 7.45 | 6.86 |
| O# | 33.94 | 39.85 | 42.95 | 41.66 |
| S | 1.94 | 1.89 | 1.76 | 1.64 |
| **HHV (MJ/Kg) db** | | | | |
| | 20.08 | 18.65 | 18.07 | 17.12 |

*Pereira et al. [18].

db: Dry basis.

%O = %C-%H-%N-%S-%H$_2$O.

photoinhibition and photooxidative damage [31]. It shows that mixotrophic cultures could be a feasible economic alternative to industrial processes.

A reduction of carbon content was observed for all biomass obtained under mixotrophic conditions when compared to autotrophic biomass (A). The reduction was more pronounced for M5 reaching 11.46%. Similarly, the same tendency was observed for nitrogen and hydrogen contents, in contrast to the oxygen content, which was higher for mixotrophic samples. The reduction of these elements in the biomass is intrinsically related to its biochemical composition, thus, there was an increase in the carbohydrate content and reduction in the protein and lipid content. Under mixotrophic conditions, microalgae can produce biomolecules different from those observed under autotrophic conditions, thus in this study there was a formation of compounds that presented lower C, N and H elemental content as well as higher O elemental content, proportionally.

With regard to carbohydrates, except for M2.5, there was a significant increase, which is expected, considering that under stress conditions, microalgae accumulate more carbohydrates as a reserve source [19,32]. Thus, an increase in the elemental oxygen content for the sample is expected once that carbohydrates are formed by C, H and O [33].

The lipids content decreased for all mixotrophic samples when compared to control being more pronounced for M2.5 and M5. The light intensity used in *S.platensis* crops for biomass production used in this study was high (238 μmol m$^{-2}$ s$^{-1}$) in order to obtain higher biomass yields. According to Yu *et al.* [34], the synthesis of lipids by microalgae can be favored under high illumination rate, however if the intensity is too high, photo-saturation may occur in mixotrophic crops. In this case, cell photosystems become inefficient or disabled, and chlorophyll molecules that are responsible for capturing light are transformed into unstable forms that react with dissolved oxygen and form reactive oxygen species that can react with free fatty

acids to disable lipid peroxidase, reducing lipid production. In this case, the decrease in lipid synthesis in mixotrophic conditions may have been caused by photo-saturation [34].

The reduction of elemental N observed in mixotrophic cultures is justified by the decrease in protein synthesis. For protein content, a reduction was observed as the concentration of cheese whey in the growing medium increased. The lowest protein content was observed for M10 that showed a reduction of approximately 40% in relation to A. These results are in agreement with the study conducted by Salla *et al.* [19,32] who demonstrated that under mixotrophic conditions, *S.platensis* increased carbohydrate synthesis and decreased protein synthesis.

In this study, nitrogen was available for microalgae from the autotrophic medium (Table 1). The organic compound assimilated by the *S.platenis* during mixotrophic growth was lactose, the source of organic carbon presented in greater quantity in the cheese whey. Cheese whey is a cloudy residue that has a high protein content and that needs to be clarified in order to be used to grow microalgae. In this clarification process, some steps are performed at high temperatures which causes the coagulation of proteins decreasing the availability of these biomolecules in the culture medium.

The oxygen content increased as the whey concentration in the culture medium was increased, being 33.94% for A and 39.85%, 42.95% and 41.66% for M2.5, M5 and M10, respectively. Therefore, there was also a reduction in calorific value, being the lowest value 17.12 MJ/Kg for M10, sample that showed the highest carbohydrates content. It can be justified by the difference in biomass composition that influences the application processes of biofuels production. The calorific values of *S. platensis* cultivated under mixotrophic conditions in this study were similar to the calorific value of wood and various lignocellulosic biomass [35,36].

There was also a reduction in sulfur content in M2.5, M5 e M10 samples. It was observed that as the whey concentration was increased in culture medium, the sulfur content decreased in biomass, being the highest reduction associated to M10 (15.5%).

Mixotrophic growth could be a microalgae cultivation mode for biomass production with lower nitrogen and sulfur content. Consequently, it also can reduce these elements in the formed biofuel, depending on the chosen process route. This strategy promotes a lower environmental impact being able to reduce the amount of S and N oxides released into the atmosphere due to burning of fossil fuels, minimizing the incidence of acid rain.

The CHNOS content of A sample was similar to that found by Anand *et al.* [37]. Elemental analysis of biomass is an important parameter since it is directly associated with the energy potential of the biomass, since high carbon content in relation to the oxygen and hydrogen content increase the biomass calorific value [38,39]. According to Bui *et al.* [40] and Figueira *et al.* [41], microalgae biomass generally has a higher carbon content, as well as a lower concentration of oxygen than traditional crop biomass. Due to the high amount of energy that is released when the C-C bonds are broken the increase in the carbon content corresponds to the increase in the fuel energy content [42]. Recent studies show that microalgae are potential biomass to produce a more stable bio-oil with better properties when compared to lignocellulosic biomass [20], however with a high nitrogen content. Jamilatun *et al.* [43] studied the bio-oil composition of *S. platensis* biomass and residue after oil extraction and observed that the elemental C content and the calorific value of both biomass and residue of *S. platensis* were similar to crude oil and wood bio-oil.

The composition of the biomass is an important factor for the design of a pyrolysis reactor. Microalgae are versatile microorganisms that can grow under different culture mediums capable of prioritizing the synthesis of one metabolite over another. Therefore, when mixotrophic culturing systems are used, there is the possibility of increasing biomass production and carbohydrate synthesis and decreasing the protein synthesis [18]. This information is important

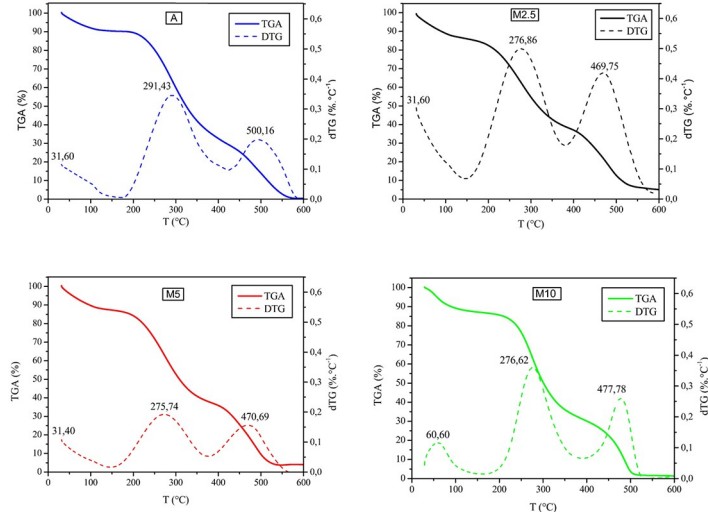

**Fig 2. TGA/DTG curves of biomass *S. platensis* degradation grown under different culture medium (a) A (b) M2.5 (c) M5 (d) M10.**

since it directly influences the composition and yield of the biofuel and, consequently, the cost of production. All these factors need to be considered for a pyrolysis process design.

## TGA of *S. Platensis*

The devolatilization of A, M2.5, M5 and M10 were evaluated, the TG/DTG analyzes were performed at a heating rate of 5°C min$^{-1}$, inert atmosphere and temperature range from 25 to 600°C. The 5°C min$^{-1}$ ratio was chosen based on the detection influence of the intermediate compounds, since lower heating ratios present a higher detection level [44]. The results obtained are shown in Fig 2.

Fig 3 shows with more detail the TGA curves referring to A, M2.5, M5 and M10 at 5°Cmin$^{-1}$. The curves show similar mass degradation profiles for all evaluated samples. The mass

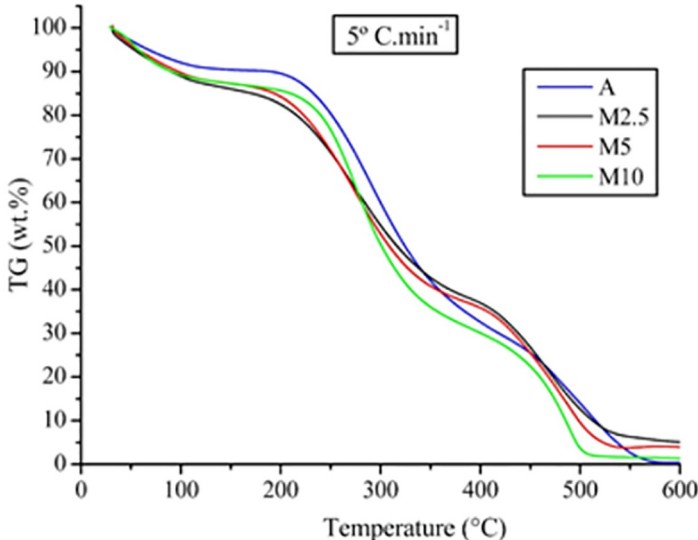

**Fig 3. TGA curves at 5 °C/min of *S. platensis* cultivated under autotrophic and mixotrophic conditions.**

**Table 3. TGA data of *S. platensis* biomass cultivated under autotrophic and mixotrophic conditions.**

| *S. platensis* | Moisture (%) | Volatile total (%) | Residue (T > 600˚C) (%) |
|---|---|---|---|
| A | 6.70 | 86.16 | 4.62 |
| M2.5 | 8.93 | 71.57 | 15.03 |
| M5 | 8.90 | 77.93 | 10.23 |
| M10 | 9.56 | 83.37 | 3.89 |

losses were maintained at similar temperatures and occurred in three stages; being the second, the most significant since it refers to decomposition of main components of *S. platensis* including carbohydrates, proteins and lipids.

The thermal degradation process of microalgae biomass occurs in three steps, as shown in Fig 2. The first step of TGA curve corresponds to the dehydration process under a range from 50 to 150˚C for A and for M2.5, M5 and M10 this range is slightly higher, ranging from 50 to 160˚C. In the DTG, the mass loss first stage for A, M2.5 and M5 presents a small peak and occurs in a range from 50 to 150˚C. For M10, the first stage occurs from 50 to 160 ˚C and presents a more defined peak represented by the DTG curve that represents 5.08% of moisture. For A, M2.5 and M5, the moisture contents were 6.35%, 8.23% and 9.43%, respectively. These values are consistent with those obtained by gravimetric analysis (Table 3), except for M10. The dehydration process corresponds mainly to the loss of water molecules and volatile compounds [45,46], at this stage internal rearrangements such as break of bonds, appearance of free radicals and formation of carbonyl (CO =) and carboxyl groups also take place (COOH) [47]. Additionally, according to Wang and Brown [48], degradation of photosynthetic pigments, including chlorophylls A and B can also occur from 80 to 120ºC.

The second step corresponds to the decomposition, presenting the largest mass losses being divided into two stages. The first one occurs the devolatilization of carbohydrates and proteins and the second presents the devolatilization of lipids and decomposition of mineral matter [41]. At this step, there are multiple peaks in the DTG curve (Fig 2) correlated with the degradation of macromolecules including proteins, carbohydrates and lipids occurring at different temperatures [41]. According to Bui *et al.* [49], normally the peaks of the DTG curve can merge, not showing the two spikes in DTG that correspond to the mass loss of carbohydrate and protein macromolecules. The mass losses, in this first stage, were A (55.56%), M2.5 (41.65%), M5 (46.23%) and M10 (53.51%) occurring from 150ºC ˚C to 400˚C (sample A), 150˚C to 375˚C (sample M2.5), 160˚C to 380˚C (sample M5) and M10 the temperature range varied from 160 to 380˚C.

A reduction in the proteins and carbohydrates was observed for M2.5 when compared to A as it can be visualized through the reduction of the total mass loss. The samples M5 and M10 showed higher carbohydrate content and lower protein content than A sample confirmed by TGA (Fig 2).

The second stage of mass loss of sample A occurred between 400ºC and 580˚C representing a percentage of 30.6%. For M2.5, M5 and M10, the mass losses were 29.92% (375 to 580˚C), 31.70% (375 to 560˚C) and 29.86% (380 to 540˚C), respectively, and represents the thermal degradation of lipid molecules. This can be justified through previous studies, such as, Peng *et al.* [50] that presented a lipid decomposition between 330ºC and 560˚C. Rizzo *et al.* [35] describes that in this second stage there is the degradation of lipids and other smaller molecules.

The initial temperature of third step mass loss varied according to the sample, starting at 580 °C for A and M2.5, 560 °C for M5 and 540 °C for M10. The residual content observed for A, M2.5, M5 and M10 were 4.62%, 15.03%, 10.23% and 3.89%, respectively. The highest values observed in the M2.5 and M5 samples are in agreement with those obtained by ashes analysis, 8.23% and 9.43%, respectively. These materials are probably carbonaceous residues, since nitrogen was used as gas inert in the TGA.

These findings are in agreement with Yang *et al.* [33], who presented a study using the microalgae *Chlamydomonas reinhardtii* and observed that the third mass loss occurs at, approximately, 575°C and corresponds to the decomposition of carbonaceous materials.

The TGA/DTG data are important for the design of process reactors for the production of bio-oil, since despite the study being carried out with the biomass *S.platensis*, the cultivation method totally influences the composition of the biomass, being an important consideration in the process. Table 3 summarizes the information obtained through TGA.

In general, the TGA indicated that the main stage of thermal degradation for samples A, M2.5, M5 and M10 occurred between 150ºC and 580°C representing a mass loss of approximately 86.16%, 71.57%, 77.93% and 83.37% for samples A, M2.5, M5 and M10, respectively. These results are in accordance with previous studies of the microalgae [9,21,47,50,51].

## Pyrolysis

In this study, a Py-CGMS was used to investigate the pyrolysis of *S. platensis* biomass obtained from autotrophic and mixotrophic conditions. The process pyrolysis was evaluated at 450, 550 and 650°C for each biomass since previous studies have been showed the highest bio-oil yield from microalgae at this range [10,11]. The choice of temperatures was based on previous studies for the pyrolysis of microalgae [10–12].

According to Pourkarimi *et al.* [51], the decomposition of biomass occurs in the first stage with internal rearrangements of molecules, bond breakage and formation of free radicals forming carbonyl and carboxyl groups, gases including CO and $CO_2$. In this stage also occurs the evaporation of water and volatile compounds and the formation of primary charred residue. In the second phase of the process, the organic compounds can be converted into gases, tar and secondary biochar through secondary reactions such as cracking, dehydration, and polymerization. As the temperature of the process increases, in the third stages occurs the decomposition of the long chains present in the liquid and in the residual solid, generating smaller molecules and gaseous fraction.

Based on the results of the chromatographic analysis performed by GC-MS, the main volatile compounds at higher concentration that were identified in the pyrolysis analyses performed in this study are shown in Table 4. The table with all identified compounds can be found in the S1-S3 Tables in S1 File.

Toluene, an aromatic hydrocarbon, and hydrocarbons such as heptadecane and heneicosane were produced by *S.platensis* pyrolysis, which makes this biomass attractive for production of high quality bio-oil. However, oxygenates and nitrogenates compounds were also produced.

The volatile compounds produced during pyrolysis were organized into groups, oxygenated, nitrogenates, N and O containing compounds, phenols, non-aromatic hydrocarbons and aromatic hydrocarbons as shown in Fig 4. The Tukey test was applied with a significance level of 5% (S4 Table in S1 File). All groups showed statistically significant differences in relation to the temperature change of the process and/or biomass composition.

It was observed that as the temperature increased, the synthesis of oxygenated compounds and N and O containing compounds decreased for all samples studied. The greatest reductions

**Table 4. List of the main semi-quantified compounds (% area) from pyrolysis of A, M2.5, M5, M10 at 450ºC, 550º and 650ºC.**

| RT | A | M2.5 | M5 | M10 | Compounds |
|---|---|---|---|---|---|
| | | | Yields (% area) at 450 ºC | | |
| 9.454 | 5,33 ± 0,30 | | | 7,61 ± 0,13 | Etanoic acid |
| 10.827 | 1,88 ± 0,31 | | 4,88 ± 0,28 | 10,80 ±0,60 | 1-hydroxy-2-Propanone |
| 15.883 | | 1,51 ± 0,4 | 2,51 ± 0,10 | 2,96 ± 0,13 | Pyrrole |
| 32.151 | 2,60 ± 0,23 | 5,00 ± 0,37 | 3,88 ± 0,66 | 1,46 ± 0,31 | Phenol |
| 47.579 | 3,05 ± 0,13 | 5,28 ± 0,01 | 4,43 ± 0,37 | 2,07 ± 0,37 | Indole |
| 53.123 | 9,96 ± 1,30 | | | | Heneicosane |
| 53.142 | | 4,47 ± 0,13 | 4,94 ± 0,13 | | Heptadecane |
| 60.869 | 4,06 ± 0,58 | 2,51 ± 0,07 | 2,86 ± 0,38 | 2,42 ± 0,26 | 3,7,11,15-Tetramethyl-2-hexadecen-1-ol |
| 68.690 | 9,71 ± 1,86 | | | 9,15 ± 0,37 | Pentadecanoic acid |
| 68.901 | 11,02 ±1,86 | | | | l-(+)-Ascorbic acid 2,6-dihexadecanoate |
| 77.925 | 8,04 ± 1,2 | 15,96 ±1,14 | 15,68 ±1,80 | 4,90 ± 0,81 | Hexadecanamide |
| | | | Yields (% area) at 550 ºC | | |
| 9.486 | 2,41 ± 0,11 | | | 5,95 ± 0,51 | Etanoic acid |
| 10.804 | 1,45 ± 0,08 | 0,84 ± 0,13 | 4,11 ± 0,02 | 7,40 ± 0,54 | 1-hydroxy-2-Propanone |
| 11.055 | 4,18 ± 1,00 | 4,82 ± 0,81 | 5,37 ± 0,40 | 3,60 ± 0,12 | Toluene |
| 15.873 | 2,30 ± 0,09 | 2,93 ± 0,16 | 3,89 ± 0,29 | 3,78 ± 0,21 | Pyrrole |
| 32.151 | 4,31 ± 0,54 | 5,72 ± 0,21 | 4,97 ± 0,61 | 3,32 ± 0,02 | Phenol |
| 36.413 | 2,16 ± 0,45 | | 2,15 ± 0,33 | 1,73 ± 0,02 | 4-methyl-Phenol |
| 43.527 | 1,52 ± 0,03 | 2,43 ± 010 | 1,92 ± 0,08 | 1,39 ± 0,13 | Benzenepropanenitrile |
| 47.545 | 3,87 ± 0,30 | 5,62 ± 0,17 | 4,86 ± 0,49 | 3,13 ± 0,05 | Indole |
| 53.090 | 6,41 ± 0,03 | 3,11 ± 0,66 | 2,84 ± 0,23 | 6,95 ± 0,54 | Heptadecane |
| 55.546 | 1,80 ± 0,04 | 2,65 ± 0,04 | 1,88 ± 0,02 | 1,17 ± 0,07 | 1-(cyanoacetyl)-Piperidine |
| 59.166 | 7,52 ± 0,18 | 5,78 ± 0,54 | 5,77 ± 0,13 | 5,31 ± 0,08 | 3,7,11,15-Tetramethyl-2-hexadecen-1-ol |
| 66.606 | 4,61 ± 0,98 | 7,06 ± 1,01 | 3,75 ± 0,49 | 3,10 ± 0,36 | Heptadecanenitrile |
| 68.901 | 15,23 ±0,39 | | | | l-(+)-Ascorbic acid 2,6-dihexadecanoate |
| 77.826 | 7,01 ± 1,64 | 10,72 ±1,00 | 8,16 ± 0,91 | 4,23 ± 0,06 | Hexadecanamide |
| | | | Yields (% area) at 650 ºC | | |
| 5.809 | 7,18 ± 0,43 | 9,77 ± 0,30 | | | Acetonitrile |
| 5.880 | | | | 7,41 ± 0,64 | Isobutylene epoxide |
| 10.823 | 1,47 ± 0,09 | | 3,48 ± 0,93 | 7,04 ± 0,06 | 1-hydroxy-2-Propanone |
| 11.027 | 7,79 ± 0,26 | 8,06 ± 0,52 | 7,50 ± 0,68 | 7,49 ± 0,01 | Toluene |
| 12.947 | 1,98 ± 0,00 | 2,96 ± 0,11 | 2,26 ± 0,01 | 1,73 ± 0,05 | 3-methyl-Butanenitrile |
| 15.834 | 2,24 ± 0,03 | 2,73 ± 0,01 | 4,01 ± 1,05 | 5,27 ± 0,12 | Pyrrole |
| 19.113 | 1,83 ± 0,06 | 3,24 ± 0,04 | 2,14 ± 0,07 | 1,69 ± 0,03 | 4-methyl-Pentanenitrile |
| 32.124 | 5,00 ± 0,28 | 5,58 ± 0,32 | 4,90 ± 0,05 | 4,06 ± 0,07 | Phenol |
| 36.425 | 3,4 ± 0,22 | 2,66 ± 0,05 | 2,41 ± 0,03 | 2,36 ± 0,18 | 4-methyl-Phenol |
| 43.487 | 1,62 ± 0,06 | 2,19 ± 0,13 | 1,70 ± 0,06 | 1,51 ± 0,13 | Benzenepropanenitrile |
| 47.511 | 3,76 ± 0,09 | 5,32 ± 0,28 | 4,42 ± 0,09 | 3,35 ± 0,41 | Indole |
| 53.055 | 5,35 ± 0,18 | 2,60 ± 0,99 | 2,64 ± 0,21 | 6,41 ± 0,92 | Heptadecane |
| 59.073 | 7,20 ± 0,00 | 5,09 ± 0,03 | 5,79 ± 0,48 | 5,01 ± 0,51 | 3,7,11,15-Tetramethyl-2-hexadecen-1-ol |
| 66.491 | 6,02 ± 0,78 | 7,09 ± 0,04 | 4,14 ± 1,22 | 3,43 ± 0,30 | Heptadecanenitrile |
| 68.751 | 8,90 ± 0,75 | | 5,96 ± 2,97 | 6,16 ± 0,91 | Pentadecanoic acid |
| 77.760 | 6,86 ± 2,47 | 8,78 ± 2,03 | 8,00 ± 1,28 | 1,52 ± 0,08 | Hexadecanamide |

*Retention Time (RT).

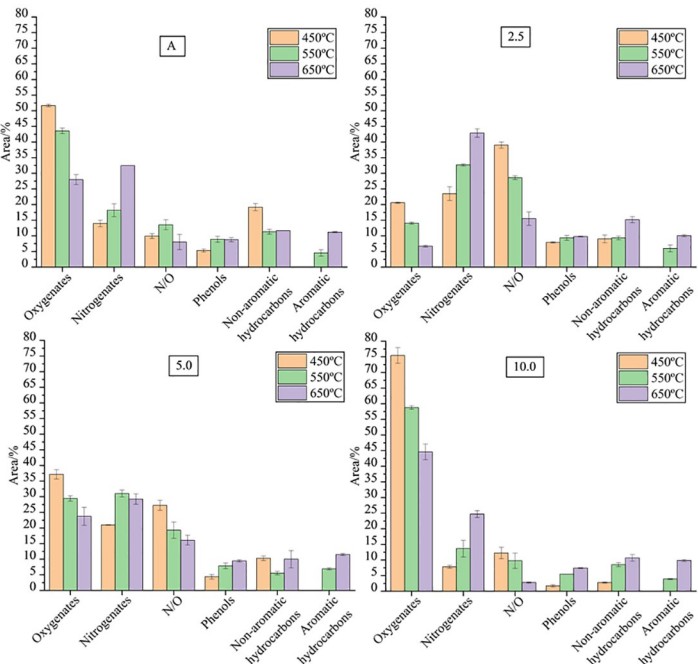

**Fig 4. Yield (% area) of volatile compounds formed by pyrolysis of *S.platensis* at 450, 550 and 650 ˚C a) A b) M2.5 c) M5 d) M10.**

in the yields of the oxygenated compounds were observed at 650ºC, the most significant were observed for A (46%) and M2.5(68%) when compared to the values obtained at 450ºC. These samples showed higher carbohydrates content than M5 and M10. This indicates that 650 ºC is the most favorable temperature for the formation of a more deoxygenated bio-oil. The main oxygenated compounds produced were etanoic acid, 1-hydroxy-2-propanone, 3,7,11,15-tetra-methyl-2-hexadecen-1-ol, pentadecanoic acid, isobutylene epoxide. During the pyrolysis process at higher temperatures, the formation of gases is favored over the production of bio-oil and biochar, thus the bio-oil produced has less oxygenated compounds, since part of the oxygen in the process is used to produce gaseous compounds [10]. Pyrolysis of M10 formed the highest percentages of oxygenated compounds representing around 75%. This was expected, since these biomasses presented higher carbohydrate contents and, consequently, higher amounts of elemental oxygen. Previous studies indicate that the oxygenated compounds formed by the pyrolysis process of microalgae are formed from carbohydrate molecules [10]. Wang and Brown [48] reported that the pyrolysis of carbohydrates forms water and oxygenated compounds and that this occurs due to the fundamental structure of monosaccharides. According to Chen *et al.* [16] during the pyrolysis of carbohydrates, primary reactions occur including dehydration, bond cleavage, ring scission, deoxygenation and rearrangement as well as the and the consequent formation of acids, aldehydes, cyclic ketones and other hydrocarbons. The carbohydrates undergo a conversion process during dehydration and deoxygenation reactions being converted to furans and olefins [52].

In general, the yields of nitrogenates were higher at higher temperatures for all evaluated biomasses. At 650˚C, nitrogenates yields were 2.4 (A), 1.8 (M2.5), 1.4 (M5) and 3.2 (M10) times greater than the value obtained at 450˚C. Simão *et al.* [11] demonstrated the content of nitrogenates compounds from pyrolysis of *Spirulina maxima* increased in response to

increases in the reaction temperature. Leng *et al*. [53] demonstrated that the N content of bio-oil increases as the pyrolysis temperature increases from 300 ˚C to 700 ˚C. They explained that the process of pyrolysis under higher temperature favors the formation of amines and amides mainly due to the decarboxylation of amino acids and the reactions between acids and $NH_3$/ $NH_2$. The content of N-heterocycles and nitriles also increases, probably due to the transformation of amines and amides into N-heterocycles and nitriles through the reactions of cyclization/dimerization and dehydrogenation/dehydration of amines/amides, respectively. Furthermore, high temperatures favor the secondary cracking of N-heterocycles into char, which contributes to the increase of N-heterocycles in bio-oil and additionally the ring-opening of N-heterocycles to produce nitriles. The formation of nitrogenates compounds also varied as a function of composition biomass. Pyrolysis of M10 showed a nitrogenates compounds reductions of 43.8% (450º), 45.6% (550ºC) and 23.8% (650ºC) compared to A. It is observed that there was a higher yield of nitrogenates in the pyrolysis of M2.5 and M5 in relation to A, however M10 showed the lowest values of all the conditions evaluated. The same behavior was observed in the formation of N and O containing compounds. About the N and O containing compounds, as the temperature increased the formation of N and O containing compounds decreased. The highest percentage of reduction was observed for M10 (77%), when the temperature increased from 450 to 650ºC and there was a reduction of 32% compared to A at 650ºC. As the percentage of whey increased in the culture medium, there was a decrease in the protein content in the samples (Table 1). In this sense, it was expected that the amount of nitrogen compounds formed during pyrolysis would show the same tendency, however, it is observed that M2.5 and M5 presented higher levels in relation to A. These results indicate that the cultivation mode for biomass productions can be optimized favoring the formation of less nitrogenates compounds by pyrolysis. However, in addition to the amount of proteins, the profile of amino acids should also influence the formation of nitrogenates products during the pyrolysis process. Thus, it is important to highlight that the formation mechanisms of volatile compounds by microalgae pyrolysis is still not very well studied. It would be necessary to know the profile of carbohydrates, proteins and lipids present in the biomass for a more accurate discussion, since it is important to evaluate not only the amount of these metabolites but also the profile of each one, as the type of molecule influences the reactive process. *S. platensis* has a high protein content, which can range from 40 to 65% just like its nitrogen content is high [54]. The nitrogen is present in the inorganic matter of the microalgae, in the form of nitrite, nitrates and ammonia, representing a percentage of 10 to 15% of the total nitrogen [50]. The most part of nitrogen in the microalgae biomass is present in amino acids forming proteins that during the pyrolysis process through reactions as of dehydrogenation, deamination, cyclization, dehydration, and dimerization [39] form the nitrogenates compounds. In this way most of the nitrogen remains in the bio-oil obtained [48]. According to Table 4, pyrrole and indole were produced for all studied conditions except for pyrolysis of A at 450ºC. In addition, acetonitrile, benzenepropanenitrile, heptadecanitrile also were the main nitrogenates compounds. At 650ºC, there was the formation of pentanenitrile, 4-methyl- and butanenitrile, 3-methyl-. Hexadecanamide was produced under all pyrolysis studied pyrolysis conditions and the highest concentrations were observed at 450ºC. It is the main nitrogen and oxygen containing compounds produced by *S.platensis* pyrolysis. The study of microalgae pyrolysis is recent when compared to the pyrolysis of lignocellulosic materials, thus the development of research in this area for the production of high quality biofuels is very relevant. The denitrogenation processes are complex and the reaction mechanisms are poorly understood [11,12]. The nitrogenates compounds obtained by pyrolysis of microalgae biomass can be useful to obtain some value-added compounds from the bio-oil that could be treated as feedstock to obtain chemicals through further treatment, rather than simply treated as fuel [55]. Depending

on the application interest, microalgae cultivation mode and pyrolysis temperature are targeted to favor a specific process route to prioritize the formation of a particular chemical of interest.

The temperature also influenced the formation of phenolic compounds mainly for sample M10 and all yields were higher at higher temperatures, being 4.3 times higher in the pyrolysis of M10 when compared to the values obtained at 450°C.These results are in agreement with previous studies. Phenol and phenol, 4-methyl compounds showed higher yields at 650°C. Vo *et al.* [56] performed the pyrolysis of the *Tetraselmis* microalgae at three different temperatures and obtained an increase in the concentration of phenolic compounds (phenol and phenol,4-methyl) with increasing temperature. Simão *et al.* [11] also demonstrated that the content of nitrogenates compounds increased in response to increases in the reaction temperature. Smaller amounts of phenolic compounds were obtained in the pyrolysis of M10, a sample that presented lower protein content. Du *et al.* [10] investigated the pyrolysis of *C. vulgaris* and albumen and phenolic compounds were produced indicating that these compounds are produced from proteins. Phenolic compounds are oxygenated and need to be separated from the bio-oil because they cause instability for applications as a fuel. However, the phenolic compounds can have various applications for production of lubricants and for manufacture of plastic, resins, automotive products, etc. [57].

In the pyrolysis of A, the yield of non-aromatic hydrocarbons decreased as the temperature increased, however the opposite happened with M2.5 and M10, which showed higher values at 650°C. In general, the yield of non-aromatic hydrocarbons was lower for M10. The most pronounced reduction was 85.5% in the pyrolysis of M10 at 450°C compared to A at the same temperature. The highest percentage was formed in the pyrolysis of A at 450°C (19.2%). Sample A showed the highest protein content, which confirm that non-aromatic hydrocarbons are formed from the decomposition of proteins by pyrolysis processes and vary with the increasing of temperature [10,38]. According to Table 4, heptadecane showed highest concentrations in the pyrolysis of M10 at 550 and 650°C. Heneicosane was formaded just in the pyrolysis of A at 450°C and the yield was 9.96 ± 1.30%. Previous studies [14,58,59] reported that the lipid content in the biomass could influence the formation of hydrocarbons, since lipids are the main precursors for the synthesis of hydrocarbons via pyrolysis.

There was no formation of aromatic hydrocarbons at 450 °C in the pyrolysis of any studied biomass. However, at 550 and 650°C aromatic hydrocarbons were formed and the highest yields were 11.2±0,2% (A), 10.0±0,3% (M2.5), 11.5±0,3% (M5) and 9.80±0,6% (M10). It is observed that there was no statistically significant difference between the yields of aromatic hydrocarbons. However, Du *et al.* [10] demonstrated that the aromatic hydrocarbons are mainly derived from the protein fraction in microalgae pyrolysis and their yields increased with temperature. They showed that toluene is formed from phenylalanine amino acid. Toluene was the main aromatic hydrocarbon produced in this study (Table 4) and it shows the potential of the *S.platensis* for the biofuel production. In addition, non-aromatic hydrocarbons are also formed through cracking, deoxygenation and deamination reactions of intermediates that occur during the protein degradation process [33].

The formation of aromatic and non-aromatic hydrocarbons by *S.platensis* pyrolysis shows the potential of this biomass to produce bio-oil and valuable chemicals. However, the formation of many nitrogenates, oxygenated and N and O containing compounds requires upgrading processes. The reduction of nitrogen in the biomass through mixotrophic growth could be a good strategy to produce biomass with less nitrogen and formed bio-oil with lower elemental nitrogen content by pyrolysis at 650°C since, as the temperature increases, the nitrogenates content increasead proportionally for samples A, M2.5, M5.0 and M10.

## Conclusions

Mixotrophic cultivation strategy of microalgae is an attractive and sustainable way to produce biomass with lower N content for production of bio-oil. In this study, the biochemical composition of *S.platensis* biomasses were different depending on the culture medium. Samples M5 and M10 showed lower protein content and higher carbohydrates content when compared to control. The different compositions of biomass influenced the elemental composition, calorific value, TGA and volatile compounds formed by Py-GC/MS at 450˚C, 550˚C and 650˚C. The main stage of thermal degradation for samples A, M2.5, M5 and M10 occurred between 150ºC and 600˚C representing a mass loss of approximately 80%. The yields of volatile compounds varied as a function of the biomass composition and temperature. In general, the yield of oxygenated, N and O containing compounds decreased as the pyrolysis temperature increased for all biomass and the lowest oxygenated yields were obtained at 650ºC for A and M2.5. Samples M5 and M10 had lower protein content and formed a smaller amount of nitrogenates compounds by pyrolysis at all temperatures evaluated. The temperature of 650ºC was the most favorable for the formation of nitrogenates compounds, phenols, aromatic hydrocarbons and non-aromatic hydrocarbons. At 650˚C, nitrogenates yield were 3.16 (M10) times greater than the value obtained at 450˚C and showed 26% of reduction compared to A. For N and O containing compounds, the highest percentage of reduction was observed for M10 (78%), when the temperature increased from 450 to 650ºC. For pyrolysis of M10 at 650ºC, the formation of phenolic compounds was 4.3 times higher compared to the values obtained at 450ºC. The highest yield of non-aromatic hydrocarbons was formed in the pyrolysis of A at 450ºC (19.2%). Aromatic hydrocarbons were formed at 550 and 650 ºC and the highest yield was observed in the pyrolysis of M5 (11.5%). These results predict that growth mixotrophic could be a strategy to decrease the N content of *S.platensis* biomass for applications in the pyrolysis process to produce bio-oil and valuable chemicals. It also demonstrated that it is possible to select temperatures and cultivation modes to favor the formation of specific chemical classes. However, it would be necessary to know the profile of carbohydrates, proteins and lipids present in the biomass for a more accurate discussion, since it is important to evaluate not only the amount of these metabolites but also the profile of each one, as the type of molecule influences the reactive process.

## Supporting information

**S1 File. Supplementary material to the manuscript.** 10.6084/m9.figshare.20278794. (DOCX)

## Acknowledgments

The authors gratefully acknowledge the Professor Dr. Carlos Henrique Ataíde (In memoriam) from the Universidade Federal de Uberlândia for pyrolysis tests, the Laboratório de Ambientes Recifais e Biotecnologia com Microalgas from the Universidade Federal da Paraíba for the support and use of their facilities and the Coordenação de Aperfeiçoamento de Pessoal de Nível Superior (CAPES).

## Author Contributions

**Conceptualization:** Sueilha F. A. Paula, Bruna M. E. Chagas, Maria I. B. Pereira, Renata M. Araújo.

**Data curation:** Sueilha F. A. Paula, Bruna M. E. Chagas, Maria I. B. Pereira, Estefani A. Asevedo.

**Formal analysis:** Sueilha F. A. Paula.

**Investigation:** Sueilha F. A. Paula, Cristiane F. C. Sassi, Renata M. Araújo.

**Methodology:** Sueilha F. A. Paula, Bruna M. E. Chagas, Maria I. B. Pereira, Adriano H. N. Rangel, Cristiane F. C. Sassi, Luiz H. F. Borba, Everaldo S. Santos, Estefani A. Asevedo, Fabiana R. A. Câmara, Renata M. Araújo.

**Supervision:** Bruna M. E. Chagas, Maria I. B. Pereira, Adriano H. N. Rangel, Cristiane F. C. Sassi, Everaldo S. Santos, Fabiana R. A. Câmara, Renata M. Araújo.

**Validation:** Sueilha F. A. Paula, Bruna M. E. Chagas, Cristiane F. C. Sassi, Luiz H. F. Borba, Everaldo S. Santos, Estefani A. Asevedo, Renata M. Araújo.

**Writing – original draft:** Sueilha F. A. Paula, Bruna M. E. Chagas, Adriano H. N. Rangel, Everaldo S. Santos, Estefani A. Asevedo, Fabiana R. A. Câmara, Renata M. Araújo.

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
