## [Decision Letter · Decision Letter 0]

26 May 2022

PONE-D-22-08124Pyrolysis-GCMS of Spirulina platensis: evaluation of biomasses cultivated under autotrophic and mixotrophic conditions.PLOS ONE

Dear Dr. Chagas,

Thank you for submitting your manuscript to PLOS ONE. After careful consideration, we feel that it has merit but does not fully meet PLOS ONE’s publication criteria as it currently stands. Therefore, we invite you to submit a revised version of the manuscript that addresses the points raised during the review process.

We look forward to receiving your revised manuscript.

Kind regards,

Ashokkumar Veeramuthu

Academic Editor

PLOS ONE

Journal Requirements:

3. Please upload a new copy of Figure 1, 3 and 4 as the detail is not clear. Please follow the link for more information: https://blogs.plos.org/plos/2019/06/looking-good-tips-for-creating-your-plos-figures-graphics/

Reviewers' comments:

Reviewer's Responses to Questions

**Comments to the Author**

1. Is the manuscript technically sound, and do the data support the conclusions?

Reviewer #1: Yes

Reviewer #2: No

Reviewer #3: Partly

2. Has the statistical analysis been performed appropriately and rigorously? 

Reviewer #1: Yes

Reviewer #2: No

Reviewer #3: N/A

3. Have the authors made all data underlying the findings in their manuscript fully available?

Reviewer #1: Yes

Reviewer #2: No

Reviewer #3: No

4. Is the manuscript presented in an intelligible fashion and written in standard English?

Reviewer #1: Yes

Reviewer #2: No

Reviewer #3: No

5. Review Comments to the Author

Reviewer #1: Dear Editor;

The manuscript entitled " Pyrolysis-GCMS of Spirulina platensis: evaluation of biomasses cultivated under autotrophic and mixotrophic conditions". It is reviewed, This study is comprehensive research. This study is interesting and useful. But there are some deficienciesis in manuscript. By this, minor revision required for article. The paper should be thoroughly revised (typing and grammar mistakes and another mistakes).The manuscript is recommended for publication.

1- Pls Explain wh did you use this material.

2- How were species selections determined?

3- You should provide more detailed information about the Pyrolysis process and characterization. The following references will be helpful. you can add

Bio-oil and bio-char from lactuca scariola: significance of catalyst and temperature for assessing yield and quality of pyrolysis.Energy Sources, Part A.

Pyrolysis of Xanthium strumarium in a fixed bed reactor: Effects of boron catalysts and pyrolysis parameters on product yields and character. Energy Sources Part A

The role of acidic, alkaline and hydrothermal pretreatment on pyrolysis of wild mustard (Sinapis arvensis) on the properties of bio-oil and bio-char. Bioresorce technology reports

3-In the conclusion section, describe the new ideas you have identified with your work.

4-The information in the sections you wrote is very scattered. Please let your explanations complement each other and be relevant and complete. The information you provide should be easy for the reader to understand.

Reviewer #2: This research provides results from the Py-GCMS of Spirulina platensis. Unfortunately, I have three major issues with this work.

The first is that this is not particularly in depth, with only a handful of experiments completed. This means there is not very much to be drawn from this study to take the field on further. This is vital to publish in a journal like PlusOne.

The second is that studies regarding the Py-GC-MS of Spirulina platensis have been published a lot elsewhere. I don't see any fundamental new insights in this work to suggest it belongs in a high impact journal such as this one.

The third is error analysis and presentation of the data. Therefore, I would suggest the authors do a little more work to try and gain a little more fundamental insight into what is happening. Do proper error analysis on the results. You simply have to have multiple repeats in Py-GC-MS work, to demonstrate that any changes in bio-crude content are not spurious.

Other comments:

Each identified compound should be given.

Please provide standard deviation for each compound rather than chemical group.

Please provide pyrograms obtained at 450, 550 and 650 ºC.

Why did these pyrolysis temperatures were chosen?

More in-depth discussions are needed during the analyses.

Reviewer #3: The manuscript "Pyrolysis-GCMS of Spirulina platensis obtained under mixotrophic growth" aims to evaluate the potential for the production of chemicals and bio-oil 131 by pyrolysis of S. platensis grown in autotrophic and mixotrophic culture medium 132 using cheese whey as a source of organic carbon by pyrolysis analysis (Py-GC/MS). It appears to be a continuation of the previous study (doi.org/10.1371/journal.pone.0224294), focusing on biomass conversion by pyrolysis.

The manuscript presents an interesting and relevant study but some points need to be considered before the final decision:

1. Abstract - the abstract does not present quantitative data, the methods need to be better described and the objective is not clear. The keyword bio-oil do not represent this study since it was not analyzed or produced. In The line 44 the authors mencioned an influence of %whey with TGA that was not clear.

2. Introduction - The main relevant/justification text of introduction can be observed between lines 122-129. As suggest, the authors could improve the introduction bringing additional information. Some paragraphs are disconnected and part of the text is not relevant to this study

Some sentences need english/langague revision for better undestand. Line 66 "In this perspective, microalgae seem to be a promising alternative biomass to produce 66 biofuel since these microorganisms offering high growth rate when compared to higher plants..." What does it mean higher plants? Line 83"... the total cost of the microalgae production process can reach up to 50% due to the inorganic nutrients and vitamins that are used..." Compare to what? Line 96 "... It has been shown that microalgae bio-oil has better chemical

97 properties than lignocellulosic bio-oil,..." and Line 116 "Pyrolysis of protein-rich biomass, including microalgae, produces a bio-oil with a high nitrogen content, reaching up to more than 10% of the elemental content, which is a major disadvantage, since this bio-oil cannot be used directly as fuel ..." seem contradictory.

3. Materials and methods

The methods should be better described.

Did the microalgae go through any preparation stage before the analysis?

Was It analyzed in fresh or lyophilized form?

What was the mass used in the pyrolysis analysis?

Line 143 - Despite the methodology being described in a previous work, it is interesting to describe the origin and characteristics of the source of nutrients in microalgae culture medium, since the objective of the work is to evaluate its influence on the products formed.

4. Results

The text between lines 195-209 do not represent a discussion of manuscript results, it describe a revision of literature.

The results should be better discussed, the text describe the results of the Table 1. Insted of describe the authors could answer some question as *Why a reduction of carbon content was observed for all biomass obtained under mixotrophic? Why lipids and protein content decreased and carbohydrates increase?

Please confirm the % residual of thermogravimetric curves (a) e (d)

Line 356 - "...hydrocarbons, oxygenated, nitrogenated, nitrogenated and oxygenated and phenolic" Does the results present in Fig 3 including the same compounds in different classes?

Line 375 - "During the pyrolysis process at higher temperatures, the formation of gases is favored over the production of bio-oil and biochar then possibly the bio-oil produced has less oxygenated compounds," Avoid o term "possibly"

Line 378 - "Previous studies indicate that the oxygenated compounds formed by the pyrolysis process of microalgae are formed from carbohydrate molecules." Please indicate the reference

Line 391 - "The formation of nitrogenated compounds varied as a function of composition biomass and temperature and the highest yields were observed at 650°C independent of biomass (Fig.4)" Why the pyrolysis high temperature increase %N compounds?

Line 403 - The whey insertion into microalgae medium improve the N supply, since it is a protein souce. The authors mencioned " If the nitrogen supply is abundant in the culture medium, there is a tendency to increase the concentrations of proteins and chlorophylls in the cells." but that was not the expected result found in the manuscript.

If Fig 3 and 4 show the same results It could be better choose one of the to present and avoid repetition.

6. PLOS authors have the option to publish the peer review history of their article (what does this mean?). If published, this will include your full peer review and any attached files.

Reviewer #1: No

Reviewer #2: No

Reviewer #3: No

---

## [Author Response · Author response to Decision Letter 0]

18 Jul 2022

The authors thank the reviewer for the considerations. The research group has been working hard to get the work published in the journal.

Reviewer #1: 

The manuscript entitled " Pyrolysis-GCMS of Spirulina platensis: evaluation of biomasses cultivated under autotrophic and mixotrophic conditions". It is reviewed, This study is comprehensive research. This study is interesting and useful. But there are some deficiencies in manuscript. By this, minor revision required for article. The paper should be thoroughly revised (typing and grammar mistakes and another mistakes).The manuscript is recommended for publication.

1- Pls Explain why did you use this material.

Response:Currently, there are challenges associated with the properties of most bio-oils that make it unsuitable for use as fuel. Bio-oil produced from lignocellulosic biomass is acidic, unstable, viscous and can contain high amounts of oxygen, dissolved water, and solids. Some studies show that bio-oil produced from microalgae and other proteinaceous biomass have better properties than lignocellulosic biomass based bio-oil. The improved properties of bio-oil obtained from the pyrolysis of a proteinaceous feedstock is associated with the conversion of its components to compounds present in the resulting oil. For example, linear hydrocarbons present in microalgae bio-oil result from the pyrolysis of lipids initially present in the algae. Microalgae bio-oil is produced not only from triglyceride conversion, but also from the conversion of proteins and carbohydrates. In fact, in the pyrolysis of algae the aromatic hydrocarbons are mainly derived from the protein fraction. 

2 - How were species selections determined?

Response: S.platensis could be a potential feedstock for pyrolysis due to its low lipid content and high protein content. Also, low-lipid algal species, as well as S.platensis, adapt easier to cultivation conditions and can be grown in wastewater with faster growth rates than high-lipid algae.

S.platensis is a species that has been highly involved in thermal degradation processes and it has important characteristics compared to others microalgae such as the high productivity of biomass and consolidated production, it can be cultivated in agricultural and industrial effluents and has an advantageous cellular structure that facilitates the harvest. In addition, this biomass has a significant carbon content and lower oxygen value when compared to a traditional lignocellulosic biomass.

3- You should provide more detailed information about the Pyrolysis process and characterization. 

Response: We agree with this and have incorporated your suggestion throughout the material and methods of the manuscript.

4-In the conclusion section, describe the new ideas you have identified with your work.

5-The information in the sections you wrote is very scattered. Please let your explanations complement each other and be relevant and complete. The information you provide should be easy for the reader to understand.

Response: The entire manuscript was revised and part of the text has been rewritten and rearranged as per the reviewer’s suggestion. 

Reviewer #2: 

This research provides results from the Py-GCMS of Spirulina platensis. Unfortunately, I have three major issues with this work.

The first is that this is not particularly in depth, with only a handful of experiments completed. This means there is not very much to be drawn from this study to take the field on further. This is vital to publish in a journal like PlusOne.

Response: Microalgae has been presented as a promising alternative biomass to produce biofuel and CO2 capture, since they offer high growth rate when compared to higher plants, cultivation versatility and adaptation to climatic variations, CO2 consumption through photosynthesis minimizing the greenhouse effect. In addition, the cultivation of microalgae can be carried out using waste from agro-industry, urban sewage, brackish and/or saline water, among others. Microalgae biomass are important to produce biofuel including bio-oil. However, the high cost of producing biomass limits large-scale applications. Mixotrophic cultivation is a preferable microalgae cultivation technique for biomass production. In this condition, biomass yields can reach 5–15 g/L, which could be 3–30 times higher than those produced under autotrophic growth conditions. Despite the advantages, pyrolysis of protein-rich biomass, including microalgae, produces a bio-oil with a high nitrogen content, reaching up to more than 10% of elemental content. This limitation compromises the industrial pyrolysis process since this bio-oil cannot be used directly as fuel. One of the ways to produce microalgae bio-oil with less nitrogen content can be via a biomass with less protein content, once that the bio-oil composition follows the trend of the biomass elemental composition. It has been shown that microalgae under mixotrophic growth conditions reduce protein synthesis and produce more carbohydrates. Thus, mixotrophic growth can be a sustainable strategy to produce S.platensis biomass with a lower protein content and, consequently, form a smaller amount of N-containing compounds during pyrolysis. In this sense, the aim of this study was to evaluate S. platensis biomass growth under autotrophic and mixotrophic culture conditions using Zarrouk medium supplemented with cheese whey. The biomasses were characterized and evaluated by thermogravimetric analysis to study the energetic potential. Furthermore, the pyrolysis of biomass was studied by Py-GC/MS at 450, 550 and 650 °C (24 pyrolysis trials) for identifying the main volatile compounds produced. It is important to mention, there is very little information available in the literature about thermal analysis studies of microalgae biomass grown under mixotrophic conditions since, which is an important parameter that will influence the design and cost of the mass degradation project for microalgae. For this study, we have a lot of data that brings answers to the evaluation of the pyrolysis of S.platensis with different biochemical composition.

The second is that studies regarding the Py-GC-MS of Spirulina platensis have been published a lot elsewhere. I don't see any fundamental new insights in this work to suggest it belongs in a high impact journal such as this one.

Yes, we can find a lot of work with S. platensis pyrolysis and this shows us how relevant this biomass is. Compared to other microorganisms, microalgae is still poorly studied and the quality and quantity of produced biomass depends on physics conditions and nutrients available in the culture medium. The same species may produce biomass with different biochemical composition that will influence the pyrolysis products. Microalgae have a great potential to produce biofuels, but the cost is still too high mainly due to the cost of biomass production.Mixotrophic cultivation has been pointed as microalgae cultivation mode for biomass/bioenergy production with lower cost. The use of whey as a supplement in culture medium for the growth of S. platensis is a good way to increase the biomass productivity and decrease the protein content. The proposals of this work were to evaluated S. platensis biomass growth in autotrophic and mixotrophic medium using whey as source of organic carbon and investigate the thermal behavior of obtained biomass by means of thermogravimetric analysis and pyrolysis (Py-GC/MS). These data are important to the projection of design, operation and modeling of thermochemical conversion systems for microalgae. 

The third is error analysis and presentation of the data. 

Therefore, I would suggest the authors do a little more work to try and gain a little more fundamental insight into what is happening. 

Response: The entire manuscript was revised and part of the text has been rewritten and rearranged as per the reviewer’s suggestion. The discussion was improved.

Do proper error analysis on the results. You simply have to have multiple repeats in Py-GC-MS work, to demonstrate that any changes in bio-crude content are not spurious.

Response: Experimental results were evaluated by Tukey's test, with a significance level of 5% (All data are available in Supporting Information S4 Table).

Other comments:

Each identified compound should be given.

Response: We have 24 pyrolysis tests and each experiment shows a lot of compounds. We have included Table 4 in the manuscript showing the main compounds produced in pyrolysis and all compounds produced are shown in the Supporting Information as per the reviewer’s suggestion (S1, S2, S3 Table).

Please provide standard deviation for each compound rather than chemical group.

Response: It has been done as per the reviewer’s suggestion. 

Please provide pyrograms obtained at 450, 550 and 650 ºC.

Response: It has been done. We have included all pyrograms in Supporting Information as per the reviewer’s suggestion (A1, A2, A3, A4, A5, A6, A7, A8, A9, A10, A11, A12, A13, A14, A15, A16, A17, A18, A19, A20, A21, A22, A23 and A24 Figure).

Why did these pyrolysis temperatures were chosen?

Response: The pyrolysis of S.platensis was performed using three pyrolysis temperatures (450ºC, 550ºC, 650 ºC). The choice of temperatures was based on previous studies involving pyrolysis of microalgae:

Z. Du et al., “Bioresource Technology Catalytic pyrolysis of microalgae and their three major components : Carbohydrates , proteins , and lipids,” Bioresour. Technol., vol. 130, pp. 777–782, 2013, doi: 10.1016/j.biortech.2012.12.115.

B. M. E. Chagas et al., “Catalytic pyrolysis-GC/MS of Spirulina: Evaluation of a highly proteinaceous biomass source for production of fuels and chemicals,” Fuel, vol. 179, pp. 124–134, 2016, doi: 10.1016/j.fuel.2016.03.076.

B. L. Simão, J. A. Santana, B. M. E. Chagas, C. R. Cardoso, and C. H. Ataíde, “Pyrolysis of Spirulina maxima : Kinetic modeling and selectivity for aromatic hydrocarbons,” Algal Res., vol. 32, no. April, pp. 221–232, 2018, doi: 10.1016/j.algal.2018.04.007.

More in-depth discussions are needed during the analyses.

Response: The entire manuscript has been revised and part of the discussion has been rewritten as per the reviewer’s suggestion.

Reviewer #3: 

The manuscript "Pyrolysis-GCMS of Spirulina platensis obtained under mixotrophic growth" aims to evaluate the potential for the production of chemicals and bio-oil 131 by pyrolysis of S. platensis grown in autotrophic and mixotrophic culture medium 132 using cheese whey as a source of organic carbon by pyrolysis analysis (Py-GC/MS). It appears to be a continuation of the previous study (doi.org/10.1371/journal.pone.0224294), focusing on biomass conversion by pyrolysis.

Response: Yes, It is a continuation of the previous study (doi.org/10.1371/journal.pone.0224294), focusing on biomass conversion by pyrolysis. 

The manuscript presents an interesting and relevant study but some points need to be considered before the final decision:

1. Abstract - the abstract does not present quantitative data, the methods need to be better described and the objective is not clear. 

Response: The abstract has been revised as per reviewer’s suggestion and the quantitative data has been added to the text and the methods have been better described as per the reviewer’s suggestion.

The keyword bio-oil do not represent this study since it was not analyzed or produced. 

Response: The bio-oil word was removed from the text as per reviewer’s suggestion.

In The line 44 the authors mencioned an influence of %whey with TGA that was not clear.

Response: All text has been revised and rewritten. This has been clarified as per reviewer’s suggestion.

2. Introduction - The main relevant/justification text of introduction can be observed between lines 122-129. As suggest, the authors could improve the introduction bringing additional information. Some paragraphs are disconnected and part of the text is not relevant to this study

Response: All text of introduction has been revised and rewritten with additional information as per reviewer’s suggestion.

Some sentences need english/langague revision for better undestand. Line 66 "In this perspective, microalgae seem to be a promising alternative biomass to produce 66 biofuel since these microorganisms offering high growth rate when compared to higher plants..." What does it mean higher plants? 

Response: Higher plants, also known as vascular plants, are a large group of plants that have vascular tissues (with veins) to distribute resources through the plant.

Line 83"... the total cost of the microalgae production process can reach up to 50% due to the inorganic nutrients and vitamins that are used..." Compare to what?

Response: The text was revised and rewritten. This has been clarified as per reviewer’s suggestion.

“According to Xia & Murphy [7], inorganic growth medium use can reach up to 50% of microalgae cultivation cost”.

 Line 96 "... It has been shown that microalgae bio-oil has better chemical

97 properties than lignocellulosic bio-oil,..." and Line 116 "Pyrolysis of protein-rich biomass, including microalgae, produces a bio-oil with a high nitrogen content, reaching up to more than 10% of the elemental content, which is a major disadvantage, since this bio-oil cannot be used directly as fuel ..." seem contradictory.

Response:All processes have advantages and disadvantages. Pyrolysis is a thermal degradation process of biomass capable to convert biomass into gaseous, liquid and solid biofuels and valuable chemicals. The liquid product, called bio-oil, has a high-energy content with great potential to replace diesel. The obtained results show that microalgae and proteinaceous biomass bio-oil has better chemical properties than lignocellulosic bio-oil, being more stable, presenting lower oxygen content and higher calorific value. These properties are justified by conversion of protein and lipids fraction into aromatic and linear hydrocarbons, respectively. S. platensis bio-oil produced through thermal conversion processes has been studied due to its potential. Moreover, the S. platensis is a high-protein species that has a high photosynthetic capacity and produces large amounts of biomass per unit area when compared to lignocellulosic and oilseed biomasses. Despite the advantages, pyrolysis of protein-rich biomass, including microalgae, produces a bio-oil with a high nitrogen content, reaching up to more than 10% of elemental content. This limitation compromises the industrial pyrolysis process since this bio-oil cannot be used directly as fuel due to the large amounts of NH3, HCN and NOx that could be released during combustion. However, compounds present in bio-oil such as pyrrole, pyridine, and indole could have important applications for pharmaceuticals and chemical industry. The N transformation behavior in the process is still not fully understood and studies involving strategies for the production of bio-oil with low N content are required. One of the techniques used is denitrogenation via bio-oil upgrading, however these are processes expensive and poorly understood. One of the ways to produce microalgae bio-oil with less nitrogen content can be via a biomass with less protein content, once that the bio-oil composition follows the trend of the biomass elemental composition. It has been shown that microalgae under mixotrophic growth conditions reduce protein synthesis and produce more carbohydrates. Thus, mixotrophic growth can be a sustainable strategy to produce S.platensis biomass with a lower protein content and, consequently, form a smaller amount of N-containing compounds during pyrolysis

3. Materials and methods

The methods should be better described.

Response: It has been done. Details of the methods have been described as per reviewer’s suggestion.

Did the microalgae go through any preparation stage before the analysis?

Response: No, It did not. The biomasses were collected in the stationary phase by filtration, then, the biomass was lyophilized and frozen at -20°C until the time of the chemical analysis. It has been described in the manuscript.

Was It analyzed in fresh or lyophilized form?

Response: The biomasses were lyophilized and frozen at -20°C until the time of the chemical analysis. It has been described in the manuscript.

What was the mass used in the pyrolysis analysis?

Response: 3 mg. It has been described in the manuscript.

Line 143 - Despite the methodology being described in a previous work, it is interesting to describe the origin and characteristics of the source of nutrients in microalgae culture medium, since the objective of the work is to evaluate its influence on the products formed.

Response: It has been described in the manuscript as per reviewer’s suggestion.

4. Results

The text between lines 195-209 do not represent a discussion of manuscript results, it describe a revision of literature.

Response: The lines were deleted as per reviewer’s suggestion.

The results should be better discussed, the text describe the results of the Table 1.

Response: The results and discussion were rewritten and additional information has been included as per reviewer’s suggestion.

 Instead of describe the authors could answer some question as *Why a reduction of carbon content was observed for all biomass obtained under mixotrophic? 

Response: The reduction of these elements in the biomass is intrinsically related to its biochemical composition, thus, there was an increase in the carbohydrate content and reduction in the protein and lipid content. Under mixotrophic conditions, microalgae can produce biomolecules different from those observed under autotrophic conditions, thus, in this study there was a formation of compounds that presented lower C, N and H elemental content as well as higher O elemental content, proportionally.

Why lipids and protein content decreased and carbohydrates increase?

Response: With regard to carbohydrates, except for M2.5, there was a significant increase, which is expected, considering that under stress conditions, microalgae accumulate more carbohydrates as a reserve source. Thus, an increase in the elemental oxygen content for the sample is expected once that carbohydrates are formed by C, H and O.The synthesis of lipids by microalgae can be favored under high illumination rate, however if the intensity is too high, photo-saturation may occur in mixotrophic crops. In this case, cell photosystems become inefficient or disabled, and chlorophyll molecules that are responsible for capturing light are transformed into unstable forms that react with dissolved oxygen and form reactive oxygen species that can react with free fatty acids to disable lipid peroxidase, reducing lipid production. In this case, the decrease in lipid synthesis in mixotrophic growth may have been caused by photo-saturation.

Please confirm the % residual of thermogravimetric curves (a) e (d)

Response: It has been confirmed as per reviewer’s suggestion.

Line 356 - "...hydrocarbons, oxygenated, nitrogenated, nitrogenated and oxygenated and phenolic" Does the results present in Fig 3 including the same compounds in different classes?

Response: The biomass conversion processes through pyrolysis form many compounds so individual compounds were grouped by class like “hydrocarbons, oxygenated, nitrogenated, nitrogenated and oxygenated and phenolic" In the revised manuscripts, the main compounds from pyrolysis were included. Supporting Information shows all compounds identified by pyrolysis of S.platensis (S1, S2 and S3 Table).

Line 375 - "During the pyrolysis process at higher temperatures, the formation of gases is favored over the production of bio-oil and biochar then possibly the bio-oil produced has less oxygenated compounds," Avoid o term "possibly"

Response: The word “possible” has been avoided as per reviewer’s suggestion.

Line 378 - "Previous studies indicate that the oxygenated compounds formed by the pyrolysis process of microalgae are formed from carbohydrate molecules." Please indicate the reference.

Response: The reference has been added in the text as per reviewer’s suggestion.

Z. Du et al., “Bioresource Technology Catalytic pyrolysis of microalgae and their three major components : Carbohydrates , proteins , and lipids,” Bioresour. Technol., vol. 130, pp. 777–782, 2013, doi: 10.1016/j.biortech.2012.12.115.

Line 391 - "The formation of nitrogenated compounds varied as a function of composition biomass and temperature and the highest yields were observed at 650°C independent of biomass (Fig.4)" Why the pyrolysis high temperature increase %N compounds?

Response: “Leng et al. [53] demonstrated that the N content of bio-oil increases as the pyrolysis temperature increases from 300 °C to 700 °C. They explained that the process of pyrolysis under higher temperature favors the formation of amines and amides mainly due to the decarboxylation of amino acids and the reactions between acids and NH3/NH2. The content of N-heterocycles and nitriles also increases, probably due to the transformation of amines and amides into N-heterocycles and nitriles through the reactions of cyclization/dimerization and dehydrogenation/dehydration of amines/amides, respectively. Furthermore, high temperatures favors the secondary cracking of N-heterocycles into char, which contributes to the increase of N-heterocycles in bio-oil and additionally the ring-opening of N-heterocycles to produce nitriles”. This discussion was added in the manuscript. 

Line 403 - The whey insertion into microalgae medium improve the N supply, since it is a protein souce. The authors mencioned " If the nitrogen supply is abundant in the culture medium, there is a tendency to increase the concentrations of proteins and chlorophylls in the cells." but that was not the expected result found in the manuscript.

Response:The discussion has been revised and some additional information has been added to clarify this point.Pereira et al. [19] demonstrated that the content of total solids in the clarified whey was 6.77% being 0.02 % of fat, 0.60 % of protein and 5.07% of lactose. In this study, inorganic nitrogen was available from the autotrophic medium (Table 1). The organic compound assimilated by the S.platenis during mixotrophic growth was lactose, the source of organic carbon presented in greater quantity in the cheese whey. Cheese whey is a cloudy residue that has a high protein content and that needs to be clarified in order to be used to grow microalgae. In this clarification process, some steps are performed at high temperatures which causes the coagulation of proteins decreasing the availability of these biomolecules in the culture medium.

If Fig 3 and 4 show the same results It could be better choose one of the to present and avoid repetition

Response: We agree with your suggestion and Fig. 4 has been removed from the manuscript.

---

## [Decision Letter · Decision Letter 1]

5 Oct 2022

Pyrolysis-GCMS of Spirulina platensis: evaluation of biomasses cultivated under autotrophic and mixotrophic conditions.

PONE-D-22-08124R1

Dear Dr. Bruna Maria E. Chagas,

We’re pleased to inform you that your manuscript has been judged scientifically suitable for publication and will be formally accepted for publication once it meets all outstanding technical requirements.

Kind regards,

Md. Asraful Alam, PhD

Academic Editor

PLOS ONE

Additional Editor Comments (optional):

Reviewers' comments:

Reviewer's Responses to Questions

**Comments to the Author**

1. If the authors have adequately addressed your comments raised in a previous round of review and you feel that this manuscript is now acceptable for publication, you may indicate that here to bypass the “Comments to the Author” section, enter your conflict of interest statement in the “Confidential to Editor” section, and submit your "Accept" recommendation.

Reviewer #1: All comments have been addressed

Reviewer #2: All comments have been addressed

2. Is the manuscript technically sound, and do the data support the conclusions?

Reviewer #1: Yes

Reviewer #2: No

3. Has the statistical analysis been performed appropriately and rigorously? 

Reviewer #1: I Don't Know

Reviewer #2: No

4. Have the authors made all data underlying the findings in their manuscript fully available?

Reviewer #1: Yes

Reviewer #2: Yes

5. Is the manuscript presented in an intelligible fashion and written in standard English?

Reviewer #1: Yes

Reviewer #2: Yes

6. Review Comments to the Author

Reviewer #1: Although necessary changes are not fully made, they are acceptable.Although necessary changes are not fully made, they are acceptable.Although necessary changes are not fully made, they are acceptable.

Reviewer #2: The authors have revised the manuscript thoroughly and carefully based on all the comments indicated by the reviewer. I recommend this paper for publication in its current form.

7. PLOS authors have the option to publish the peer review history of their article (what does this mean?). If published, this will include your full peer review and any attached files.

Reviewer #1: No

Reviewer #2: No

---

## [Editor Report · Acceptance letter]

11 Oct 2022

PONE-D-22-08124R1 

Pyrolysis-GCMS of *Spirulina platensis*: evaluation of biomasses cultivated under autotrophic and mixotrophic conditions 

Dear Dr. Chagas:

I'm pleased to inform you that your manuscript has been deemed suitable for publication in PLOS ONE. Congratulations! Your manuscript is now with our production department. 

Kind regards, 

on behalf of

Dr. Md. Asraful Alam 

Academic Editor

PLOS ONE